# Household coverage, quality and costs of care provided by community health worker teams and the determining factors: findings from a mixed methods study in South Africa

Jane Goudge ,[1] Julia de Kadt,[1] Olukemi Babalola,[1] Michel Muteba,[2] Yu-hwei Tseng,[1] Hlologelo Malatji,[1] Teurai Rwafa,[1] Nonhlanhla Nxumalo,[1] Jonathan Levin,[2] Margaret Thorogood,[3] Emmanuelle Daviaud,[4] Jocelyn Watkins,[3] Frances Griffiths [3]

For numbered affiliations see end of article.

**Correspondence to**
Dr Jane Goudge;
Jane.goudge@gmail.com

## ABSTRACT

**Objective** Community health workers (CHWs) are undertaking more complex tasks as part of the move towards universal health coverage in South Africa. CHW programmes can improve access to care for vulnerable communities, but many such programmes struggle with insufficient supervision. In this paper, we assess coverage (proportion of households visited by a CHW in the past year and month), quality of care and costs of the service provided by CHW teams with differing configurations of supervisors, some based in formal clinics and some in community health posts.

**Participants** CHW, their supervisors, clinic staff, CHW clients.

**Methods** We used mixed methods (a random household survey, focus group discussions, interviews and observations of the CHW at work) to examine the performance of six CHW teams in vulnerable communities in Sedibeng, South Africa.

**Results** A CHW had visited 17% of households in the last year, and we estimated they were conducting one to two visits per day. At household registration visits, the CHW asked half of the questions required. Respondents remembered 20%–25% of the health messages that CHW delivered from a visit in the last month, and half of the respondents took the action recommended by the CHW. Training, supervision and motivation of the CHW, and collaboration with other clinic staff, were better with a senior nurse supervisor. We estimated that if CHW carried out four visits a day, coverage would increase to 30%–90% of households, suggesting that some teams need more CHW, as well as better supervision.

**Conclusion** Household coverage was low, and the service was limited. Support from the local facility was key to providing a quality service, and a senior supervisor facilitated this collaboration. Greater investment in numbers of CHW, supervisors, training and equipment is required for the potential benefits of the programme to be delivered.

### Strengths and limitations of this study

► A wide range of data collection methods enabled triangulation across data sources to provide a more accurate description of the community health worker (CHW) programme.
► The six sites were typical of the Sedibeng District, and while it is not possible to generalise with any certainty, the sites included both rural and periurban communities typical of much of South Africa.
► Due to the very low number of respondents reporting a CHW visit in the past month, analyses of disease-specific services were not possible.
► During observations, the work effort by the CHW increased due to the presence of fieldworker, although we observed the same CHW over 4 days to reduce this effect.

## INTRODUCTION

Community health worker (CHW) programmes have the potential to improve access to care for vulnerable communities,[1 2] and they can be effective in improving health behaviours and outcomes.[3 4] However, many programmes do not have adequate supervision and resources, resulting in low motivation and poor performance among the CHW, and the expected health benefits do not materialise.[5] With the increasing importance of universal healthcare coverage, CHWs are responsible for a greater range of promotive and preventative care, the skills required are broader and more complex, and the need for supervision is greater.[6] However, due to the shortage of healthcare workers in low-income and middle-income settings, the number of nurses available to supervise CHW is limited.[7 8] As a result, programme

BMJ

managers often have to choose between locating CHW teams in established clinics, with closer supervision and greater access to resources, or locating teams in health posts away from the clinic but closer to communities in need, enabling a more responsive service and reducing CHW travelling.

South Africa initiated a national CHW programme (called ward-based outreach teams) in 2011 to strengthen primary healthcare.[9] The intention is to provide health promotion, prevention, screening services and referral for a wide range of health conditions.[10] While policy documents suggested that CHW should care for approximately 150–250 households,[11] it is not clear how many households the CHW are able to serve, or whether they are able cope with the broader range of health needs they are meant to provide services for. In deciding how to strengthen and to scale up this programme further, decision-makers need information on the household coverage and quality of care achieved by differing supervisor/location models, as well as the varying costs from the existing teams.

In the initial observation phase of a 3-year intervention study in Sedibeng Health District, we studied CHW teams with different configurations of supervisors and locations: clinic-based teams supervised by a professional nurse (PN) and an enrolled (junior) nurse (EN); health post-based teams supervised by a PN and an EN; and clinic-based teams supervised by an EN only. In this paper, we report on household coverage, quality and costs of the service of the different models.

## METHODS
We used a case study approach,[12] combining qualitative and quantitative data, to examine the operation of six CHW teams, with each team and their supervisors being a single case study.

### Study site
In the South African government CHW programme, CHW teams are meant to comprise a nurse, six or more CHWs, one health promoter and one environmental officer.[9] In Sedibeng health district (Gauteng Province), at the time of the study, there were 39 CHW teams in 37 of the district's 72 wards (smallest geopolitical area). The teams varied in level of supervision and location of the team's base.

The supervisors were either professional or enrolled nurses. Professional nurses (PNs) in South Africa are able to diagnose patients, prescribe treatment and dispense medication. The PN supervisors were also trained in primary healthcare and community nursing. Enrolled nurses (ENs) complete a 2-year nursing course and are qualified to provide nursing care under supervision.

Sixteen of the teams were based at a health post and the remaining 23 were clinic-based. Some health posts were located relatively close to their 'mother' clinic with the aim of providing outreach services; other health posts

**Table 1** Configurations of CHW teams

| Model | Supervisor | Based in | Site number |
|---|---|---|---|
| 1 | Professional and enrolled nurse | Clinic | 1 |
| | | | 2 |
| 2 | Professional and enrolled nurse | Health post | 3 |
| | | | 4 |
| 3 | Enrolled nurse only | Clinic | 5 |
| | | | 6 |

were located in communities without a clinic in order to improve access to basic services. A health post consists of one or two temporary wooden structures (providing three to six rooms), without electricity and often with irregular water supply. It is managed by one or two PNs, who obtain medication and other resources via a 'mother' clinic. The nurses supervise the CHW team and provide basic services to the community such as chronic medication, immunisation and treat minor ailments.

Nation-wide standardised CHW training covers identification of the need for antenatal and postnatal care, monitoring immunisation and adherence to chronic medication, screening for malnutrition, tuberculosis, gender-based violence, and making referrals to health and social services. Registration of a household includes questions about coughing, HIV testing, pregnancy, recent births, children under 5, chronic medication and need for family planning, home-based care, social grants. In addition, in Sedibeng, the CHWs deliver chronic medication to elderly, or disabled, patients.

We categorised the CHW teams into three types and selected two teams of each type (table 1). We drew on the advice of the district managers, to select each pair of teams to maximise the similarity in the community served by the two teams.

### Data collection
In each site, we conducted a household survey, focus group discussions (FGDs) with the CHW, observations of CHWs at work and interviews with CHWs' clients, CHW supervisors, facility managers and the community representatives from September 2016 to February 2017 (table 2).

#### Household survey
We used stratified sampling (based on area and housing type)[13–15] to select 220 households per site. Fieldworkers used a random walk and a specified skip pattern in a designated area to select households. The household was approached and the member who knew most about the health of other members was invited to participate in the survey. Their responses were recorded on an electronic device. This allowed any irregularities to be identified and resolved as the survey progressed. We collected data on coverage (whether a household had been visited in the past year or in the past month), the need for care (number and types of health conditions) and the

**Table 2** Data collection method, participants and data collected

| Component | Participants | Total number in six sites | Data collected |
|---|---|---|---|
| Quantitative | | | |
| Household survey | Randomly selected households in the catchment area (220 per site) | 1227 household interviews | ► Socioeconomic status of household, demographic profile, access to care and types of need for care<br>► Whether a CHW had visited in the past month/year (coverage)<br>► Health messages recalled by the HH (as a proxy for quality of care) |
| Costs | Setup and recurrent costs per site | – | Setup and recurrent costs |
| Qualitative | | | |
| FDG | CHW teams | 12 FGDs (76 participants) | ► Types of activities, challenges of the programme<br>► Self-administered questionnaire: age, years of training and service |
| Observations | CHWs and supervisors while conducting their daily work | 126 days of observation | ► Types of activities carried out; types of clients encountered<br>► Supervision activities and interaction with clinic staff |
| In-depth interviews | Supervisors, facility managers, clinic staff, community key informants | 43 key informant interviews | Perceptions of what the programme entailed, how it ran, its successes and challenges |
| | Follow-up interviews with CHW clients who were referred to the clinic during observations | 74 household interviews | Whether client took referral action, and outcome client's perception of the service |

CHWs, community health workers; FGD, focus group discussion.

quality of care provided (the number of health messages recalled for visits in the last month). Health messages are predefined advice concerning each condition that the CHW was meant to provide, when relevant. Descriptive statistics were generated for all variables of interest and compared across sites. A logistic regression was conducted to better understand who was receiving CHW services. We included the following variables: model type, ratio of households per CHW pair, whether the household has a person over 60, a child under 5, number of healthcare needs, distance from the facility and dwelling type (as a proxy for socioeconomic status (SES)). We first conducted an unadjusted analysis to determine whether any of the variables were significantly associated ($p<0.2$) with receiving a CHW visit. All significant explanatory predictors were included in the model, with those that were not significant ($p>0.05$) being removed one by one, with the least significant being removed first. The analysis took into account the effect of clustering due to sampling and levels of strata by using a robust SE estimates for stratified sampling.

## Qualitative data collection

Trained fieldworkers were given an orientation to community-based healthcare by an experienced primary healthcare nurse. Interview guides and observation templates were revised after internal piloting and feedback from fieldworkers. Respondents, except for CHW, were chosen purposively.

### Focus group discussions

All CHWs at work on the day of the FGD were included. A brief survey captured key demographic and career history of the CHW. The topics discussed included descriptions of the types of activities carried out by CHW and the challenges they faced.

### Observation of CHW

The selection of CHW to be observed was done randomly (drawing names out of a hat) on the first morning of a 4-day observation period. The fieldworkers observed the CHW with or without supervisor at work and took detailed notes in a template.

### Interviews

Interviews were conducted with the facility manager, clinic staff and CHW supervisors to discuss the typical activities of the CHW, how the programme ran, and its successes and challenges. If a client was given a referral by a CHW, the fieldworkers asked to conduct an interview with the client in a month's time. This subsequent interview provided information on follow-up actions taken by the client.

We drew together qualitative data to develop an explanatory description of how each team functioned, and then drew comparisons across teams. First, we extracted data for each site into one document, either summarising events or useful quotations as raw data. (Team members extracted data from same sources, compared extracted data and modified extraction strategies until we were confident about inter-extractor reliability). We presented a brief summary of each site in a 1-day workshop, and through discussion identified common and divergent themes across the sites (such as weekly and daily pattern of activities, resources, record keeping, patient referrals to the clinic, engagement with clinic staff, relationship between CHW and supervisors, relationship with patients). We then collated data under these themes, producing a summary for each site (see example in online supplementary appendix 1). This allowed us to further compare sites, to see how variation in supervision and integration with the health system influenced the functioning of the teams.

In addition, we counted the observed visits by purpose (medication delivery, follow-up visit, registration), which questions were asked during registration, and, from the interviews with clients, whether the referral action advised by the CHW was taken and reasons why not. (Registration required the completion of a government form identifying nine health conditions/needs).

### Costing

We only included the costs to the health system, not household costs, or those incurred by CHWs such as airtime or transport. We included both setup costs (health post infrastructure and equipment, CHW kits and training) and recurrent costs (staff costs, health post and equipment maintenance, CHW kit replenishment and overheads). Staff costs included CHWs, supervisors and a share of the responsible district manager. Costs and the estimated life span of infrastructure and equipment were collected from the district management team. PHC expenditure per capita for the district was extracted from the Health System Trust District Barometer 2015. All costs are expressed in 2016 Rands after applying adjustment for inflation if required.

All items with an expected life over 1 year were treated as setup costs and annualised by expected length of life, using straight depreciation. Costs were derived for each site, based on the particular site's staffing profile. Costs for the district CHW programme were calculated as expenditure per capita. This amount is compared with the per capita PHC expenditure to determine the share of PHC expenditure represented by the CHW programme.

### Integrating the qualitative and quantitative evidence

We compared the data on coverage and quality of care from the survey with the qualitative evidence in order to examine how the different models functioned and why. We then combined these judgements with the costs to make an assessment about the value for money of the different models.

### Ethics

Written consent was obtained from all participants. Verbal consent was obtained from household members for a fieldworker to enter the household to observe the CHW at work. Written consent was obtained from household members when subsequent interviewers were conducted.

### Patient and public involvement

CHW clients and the public were not involved in the design, conduct, reporting or dissemination plans; however, facility, district and provincial managers were involved. We have conducted feedback sessions with CHW teams, facility, district and provincial managers.

## RESULTS

### The clinics, health posts and the CHW teams

The clinic in Site 1 was a formal, relatively modern building (table 3). The clinic in Site 2 was prefabricated wooden structures with limited space, so the CHW had to meet outside. The health post in Site 3 (a repurposed transport container) was located relatively close to its 'mother' clinic. In contrast, the health post in Site 4 was located in a community without a clinic. As a result, at this site, the PN supervisor spent half of her time consulting with patients, rather than supervising the CHW. Sites 5 and 6 were both formal clinics in rural settings. Site 6 had sufficient space for CHW to their meetings inside; Site 5 did not and CHW met outside.

The EN supervisors (other than in Site 1) were younger and had less work experience than the CHW. There were no PN supervisors at the two rural sites (Sites 5 and 6); the district manager reported it was hard to recruit PNs to work in the rural clinics because of their location. The size of the teams ranged from 9 to 20 CHWs. More supervision time per CHW was available to the teams in Sites 1 and 2, and to a lesser extent Site 3. Using the South African census, we estimated the number of households per pair of CHW varied from 120 to 400 (table 3).

The average age of the teams varied between 33 and 41 years, and the percentage who had completed their high school education ranged from 25% to 63%. Length of time working as CHW ranged from 5 to 10 years. The majority of CHW had completed Phase 1 WBOT training (90%–100%), except for the two rural sites (Sites 5 and 6) that were located over an hour's drive from the training facility.

The CHWs often worked without necessary resources (equipment, stationery, uniform or funds for transport or mobile communication). The CHWs were meant to have a bag of equipment, which included a blood pressure machine, a glucometer and testing strips, a weighing scale, mid-upper arm circumference tape for assessing malnutrition in children, bandages, gloves, mask, forms, raincoat and water bottle. At the start of the fieldwork,

**Table 3** Description of catchment area, households and members surveyed, and their need for care

| | PN & EN based in clinic | | | | PN & EN based in health post | | | | EN only based in clinic | | | |
|---|---|---|---|---|---|---|---|---|---|---|---|---|
| | Site 1 | | Site 2 | | Site 3 | | Site 4 | | Site 5 | | Site 6 | |
| **2016 Department of Health District data** | | | | | | | | | | | | |
| Catchment area | | | | | | | | | | | | |
| Population in CHW catchment area (N) | 23 877 | | 6749 | | 10 984 | | 5000 | | 5838 | | 16 610 | |
| Household per site (N) | 6453 | | 2177 | | 3328 | | 1429 | | 1824 | | 4746 | |
| No. of people per household (Mean) | 3.7 | | 3.1 | | 3.3 | | 3.5 | | 3.2 | | 3.5 | |
| Households surveyed (n) | 187 | | 186 | | 224 | | 213 | | 209 | | 206 | |
| SES, access to, and need for, care | | | | | | | | | | | | |
| Proportion of households that were informal dwellings (%, n) | 0 | 0 | 25.8 | 48 | 14 | 30 | 96 | 205 | 43 | 90 | 24 | 47 |
| Access to piped water indoors (%, n) | 54.6 | 102 | 39 | 73 | 54 | 120 | 3.8 | 8 | 44 | 92 | 65 | 133 |
| Access to internet (yes) (%, n) | 42.3 | 79 | 25.8 | 48 | 34 | 76 | 31 | 66 | 25 | 53 | 33 | 67 |
| Walk to the clinic (yes) (%, n) | 77.4 | 144 | 63.4 | 129 | 89 | 199 | 75 | 159 | 89 | 186 | 87 | 179 |
| Distance from house to clinic/health post (km) (median, IQR) | 0.7 | 0.5–1.0 | 1.7 | 1.5–2.2 | 0.4 | 0.2–0.7 | 0.4 | 0.3–0.6 | 2.4 | 2.0–2.9 | 0.6 | 0.4–1.3 |
| Three or more health conditions or needs (%, n) | 18.2 | 34 | 12.4 | 23 | 12 | 26 | 16 | 35 | 20 | 41 | 14 | 29 |
| Individuals in households surveyed (n) | 692 | | 574 | | 747 | | 755 | | 679 | | 722 | |
| Age | | | | | | | | | | | | |
| 0–4 (%, n) | 6.5 | 45 | 7.8 | 45 | 7.9 | 59 | 3.8 | 29 | 4.3 | 29 | 7.9 | 57 |
| 5–18 (%, n) | 19.4 | 134 | 26.7 | 153 | 27 | 201 | 27 | 202 | 26 | 176 | 29 | 206 |
| 19–59 (%, n) | 58.7 | 406 | 59.6 | 342 | 59 | 437 | 62 | 465 | 63 | 428 | 56 | 405 |
| 60+ (%, n) | 15.5 | 107 | 5.9 | 34 | 6.7 | 50 | 7.8 | 59 | 6.8 | 46 | 7.5 | 54 |
| **Clinics, health posts and CHW teams** | | | | | | | | | | | | |
| Description of clinic or health post and availability of space | Modern clinic building | | Prefabricated clinic building | | Transport container, close to 'mother' clinic | | Prefabricated building some distance from 'mother' clinic | | Modern clinic building in rural setting | | Modern clinic building in rural setting | |
| Does the team have a room inside the facility to use for meetings? | Yes | | No | | Yes | | Yes | | No | | Yes | |
| No. of CHW per team | 16 | | 17 | | 9 | | 12 | | 14 | | 20 | |
| No. of households per CHW pair | 403 | | 128 | | 396 | | 119 | | 130 | | 237 | |
| Proportion of CHW who have finished high school education | 27% | | 35% | | 50% | | 63% | | 25% | | 33% | |

**Table 3**  Continued

| | PN & EN based in clinic | | PN & EN based in health post | | EN only based in clinic | |
|---|---|---|---|---|---|---|
| | Site 1 | Site 2 | Site 3 | Site 4 | Site 5 | Site 6 |
| Proportion of who have passed Phase 2 training | 90% | 24% | 100% | 45% | 0% | 0% |
| No. of professional nurse supervisors | 1 | 1 | 1 | 1 | 0 | 0 |
| No. of enrolled (junior) nurses as supervisors | 2 | 2 | 1 | 1 | 1 | 1 |

Distance, direct distance between GPS coordinates.
CHW, community health worker.

many CHWs were without bags. Equipment bags were delivered during the data collection but were not regularly restocked. In one team, the CHW had not received training on how to use the manual blood pressure machines provided. The notebook and pen that CHW used to record daily activities had to be purchased out of their stipend, and so many of them did not have these items during our observations. They used pieces of papers to record the visits. On occasions, CHWs did not have the household registration form because the photocopier was broken. Two of the six teams did not have sufficient space to complete paperwork and to store their files; often files were kept at home.

The CHWs were not formally employed by the government. They were an outsourced workforce and paid a minimal stipend for 6 hours work per day. CHWs struggled to contact the private payroll company responsible for monitoring attendance and paying stipends. An electronic system, requiring CHWs to clock in/out at the clinic, limited their ability to reach outlying areas. Two months prior to the start of our fieldwork CHWs had been on strike over their conditions of employment. During the fieldwork the CHW were paid (2500 RSA), below the minimum wage (R3500 per month); however, by the end of 2018, it was increased to the minimum wage.

### Household characteristics, need for, and access to, care
The proportion of households who agreed to participate in the survey ranged from 85% to 97% across the sites. Site 1 contained mostly formal dwellings, all of which had access to piped water inside their house (table 3). In contrast, Site 4 was predominately an informal settlement, with only 3.8% of households having piped water inside the house; in Site 5, half of the dwellings were informal. Across the six sites, access to the internet ranged from 25% to 42% of households, with 31% of households having access in the informal settlement (Site 4). The age distribution of the population is similar to the informal settlements (Sites 4 and 5) having fewer children under 5, and the more formal settlement (Site 1) having double the number of people aged 60 and over. The need for care was greatest in the informal settlements; in Site 5, 19% of households reported three different conditions

or need for care due to high levels of HIV and TB. The formal settlement (Site 1) had higher levels of hypertension and diabetes due to its elderly population.

### Household coverage
Across the sites, 10%–20% of households had been visited in the last year, with 5%–12% visited in the last month (table 4). Having only an EN supervisor was associated with a higher level of coverage. Those households with a person aged 60 and above were also almost twice as likely to receive a visit in the past year (due to the delivery of medication to elderly patients). However, other characteristics that we would expect to increase the likelihood of a visit, such as a child under 5, the number of healthcare needs and dwelling type (a proxy for SES), did not. CHWs were not more likely to visit households closer to the clinic or health post. Combining the household survey data with the estimated number of households in each catchment area, we estimated that a pair of CHW on average visited between one and two households per day. However, during the observations, the CHW visited on average of three to five households per CHW pair per day, the increase effort probably due to being observed.

### Type, and quality, of care delivered
The purpose of household visits varied. CHW delivered medication in nearly half of observed visits (47%), followed up with a patient (38%) or registered a household not visited before (15%). None of the observed household registrations were fully completed, with CHWs completing on average four to five of the nine questions (table 5). The three most frequently asked about health conditions/needs were as follows: a children under 5, a person taking daily medication and a birth in the last 6 weeks (in >70% of registrations). The need for HIV testing and family planning was established in at least half of registration visits. Coughing, the need for social grants and pregnancy status were the least discussed. In a third of visits, an additional health need was established, as CHWs asked about other health issues, although this varied across sites (72.5% in Site 4% and 18.2% in Site 3).

The survey respondents in Sites 1 and 2 (with PN and EN supervisors) recalled a quarter of the relevant

**Table 4** Household coverage and quality of care

| | PN & EN based in clinic | | PN & EN based in health post | | EN only based in clinic | |
| --- | --- | --- | --- | --- | --- | --- |
| | Site 1 | Site 2 | Site 3 | Site 4 | Site 5 | Site 6 |
| **Coverage** | | | | | | |
| All households SURVEYED (n) | 187 | 186 | 224 | 213 | 209 | 206 |
| Households visited by CHW in the last year (%, n) | 10  19 | 12  22 | 16  35 | 16  34 | 20  41 | 20  41 |
| Households visited by CHW in the last month (%, n) | 5.3  10 | 7.5  14 | 8.9  20 | 6.6  14 | 12  26 | 5.8  12 |
| **Purpose of visit, whether registration completed or additional need identified** | | | | | | |
| Household visits OBSERVED (n) | 81 | 87 | 77 | 40 | 89 | 128 |
| Household registration (%, n) | 27  22 | 6.8  6 | – | 73  29 | – | 16  21 |
| Medicine delivery (%, n) | 67  54 | 48  42 | 60  46 | 20  8 | 49  44 | 34  43 |
| Follow-up (eg, missed clinic appt, hospital discharge) (%, n) | 6.2  5 | 45  39 | 43  33 | 7.5  3 | 51  45 | 52  66 |
| Average number of registration questions asked (%, n) | 54  4.8/9 | 56  9 May | – | 50  4.5/9 | – | 44  9 April |
| Visits whereby CHW found a need (%, n) | 60  49 | 28  24 | 18  14 | 73  29 | 19  17 | 36  46 |
| **Advice recalled** | | | | | | |
| All households SURVEYED (n) | 187 | 186 | 224 | 213 | 209 | 206 |
| Proportion of relevant health messages recalled (ratio, SE) | 0.3  0.03 | 0.3  0.03 | 0.2  0 | 0.1  0.04 | 0.2  0 | 0.1  0.07 |
| Proportion of messages recalled for hypertension (ratio, SE) | 0.3  0.02 | 0.3  0.03 | 0.2  0 | 0.1  0.05 | 0.2  0 | 0.2  0.09 |
| **Referrals acted on** | | | | | | |
| Total individual interactions during OBSERVED HH visits (n) | 127 | 128 | 112 | 79 | 136 | 165 |
| No. of referrals recorded (%, n) | 13  16 | 13  17 | 17  19 | 15  12 | 19  26 | 10  17 |
| No. of referred individuals interviewed (%, n) | –  14 | –  13 | –  15 | –  6 | –  16 | –  13 |
| Patients who took referral action (%, n) | 57  8 | 54  7 | 53  8 | 50  3 | 69  11 | 39  5 |

messages for the conditions discussed during the visit in the last month (table 5). However, respondents in Sites 4 (health post team in a community without access to a clinic) and 6 (one of the EN-only sites) recalled only 11% and 13% of the relevant messages.

Half of the clients sought care at the clinic when referred by the CHW. The most common reason given for no action was lack of funds or transport. In Site 6, only 39% of referrals were acted on. Key community informants reported complaints had been made about staff attitudes at the clinic, which may have influenced whether the patient took action.

### Supervision practices and support from the health system

We observed experienced supervisors using job training, supervised household visits and debriefing sessions (during which problems were discussed) to train, motivate and monitor CHW, and improve the quality of their work. In one site, the PN gave brief education sessions about common health conditions in the community before the CHW left the clinic in the morning, '*I cover hypertension, diabetes, HIV, the children's Road to Health card, TB, health hazards, prostate and cervical cancer screening. It is things they deal with in the community.*' (PN, interview, Site 1). One health post-based PN/EN team (Site 3) held daily debriefing sessions, at which CHWs described the households visited, problems encountered and actions taken. The sessions strengthened CHWs' knowledge base and problem-solving abilities, and provided a platform for collective supervision, enabling the CHWs to learn from each other's experience. Those teams that had no place to meet inside the facility tended not to discuss serious matters for reasons of confidentiality. *"Sometimes we stand outside to hold meetings…patients and the security personnel will be listening to what we are discussing." (CHW-FGD, Site 5)*

ENs routinely accompanied different pairs of CHWs on home visits several days a week. One EN (who worked as part of an EN/PN pair) demonstrated sensitivity in correcting CHWs' practice. *"When I can see this one didn't do it right, I keep quiet in the house, but immediately after we step outside as we are walking, I do on-the-spot training." (EN, interview, Site 2)* Despite one EN (without a PN supervisor) spending 4 days a week out in the community, the CHWs

**Table 5** Annualised costs of the different models

| Rands 2016 | PN & EN based in clinic | | PN & EN based in health post | | EN only based in clinic | |
| --- | --- | --- | --- | --- | --- | --- |
| | Site 1 | Site 2 | Site 3 | Site 4 | Site 5 | Site 6 |
| No. of PN supervisors | 1 | 1 | 1 | 1 | 0 | 0 |
| No. of EN supervisors | 2 | 2 | 1 | 1 | 1 | 1 |
| Total costs per team | 1 047 953 | 1 083 809 | 938 344 | 910 240 | 694 429 | 909 564 |
| No. of CHW | 16 | 17 | 9 | 12 | 14 | 20 |
| Cost per CHW | 65 497 | 6 753 | 104 260 | 73 853 | 49 602 | 45 478 |
| Explanation for variation in costs | Two ENs in each of these sites, so total costs are higher, but the teams are larger so supervision is needed | | Fewer CHW per supervisor so more expensive, as well as the costs of the health post | Larger team so cheaper than Site 3, but one supervisor actually consulting patients | With a fewer supervisor and larger teams, the costs were lower | |
| Quality of care | High quality | | High quality | Poorer quality because less supervision and significant distance to facility | Poor quality, because insufficient supervision | |
| Value for money | Good value: good supervision, and well integrated into the health system, even in clinic where there is not enough space for the CHW | | High quality model, but intensive because few CHW, and so expensive. | In mid-range in terms of expense but trying to provide basic clinic services as well, so quality suffers | Poor value: cheaper but poor quality, ineffective care | |

CHW, community health worker; EN, enrolled nurse; PN, professional nurse.

did not acknowledge her as a supervisor, *"When we get there she's doing the same job that I normally do when I am alone." (CHW-FGD, Site 5)*

Job satisfaction, professional confidence and motivation were more evident in teams with a PN supervisor. One CHW commented that since the PN arrived: *"we started to feel that we are now CHWs and human beings." (CHW-FGD, Site 1)* When the CHWs were on strike over their conditions of employment, some of the CHWs at the clinic-based PN/EN Site (Site 2) tried to carry on, not wanting to let their patients down or damage their relationship with supervisors and the facility. Their supervisor interpreted their actions: *"they see that you, as their leader, are supporting them and trying to understand where they come from, especially with this employer thing being mixed up. That is why I am saying, hence, they refused to go on strike today." (EN, interview, Site 2)*

However, in the absence of supportive supervision in EN-only sites, insufficient training constrained CHWs' ability to assist patients: *"since we work under WBOT we have not been trained." (FGD Site 6)* Based an hour's drive from the district office, the rural sites rarely received training. With their frustrations over their unresolved working conditions, which were not acknowledged by their immediate supervisors, the CHW were demotivated: *"the way the contractor is operating, they make us to be lazy, because since we signed a contract with them, we have never seen them or had*

*a meeting with them to tell us about their rules." (FGD Site 6)* Field notes also showed that many CHW did not make full use of their time due to a lack of motivation and ineffective time-task management. Some CHWs decided to work fewer hours a day as a passive protest.

In contrast, in the clinic-based teams, with the support of PN supervisors, and the resulting growth in skills and confidence, CHWs had been able to establish better relationships with clinic staff. In turn, clinic staff, seeing the CHW's greater motivation and effort, were more likely to treat the CHWs with respect. Communication and coordination was better. Working together, the facility manager and PN were able to confirm whether, once defaulters had been traced by CHWs, they returned to the clinic. *"We engage with facility manager, she helps with problems that we have…we include her in everything." (EN, interview, Site 2)* Clinic staff willingly accepted referrals from CHWs, and included the CHWs in training sessions: *"Sister X is not (in the CHW team), but she is very helpful. She attended severe malnutrition training and she called all the CHWs and did a presentation for everybody." (EN, interview, Site 2)* As a result, the contribution of CHWs was highly valued: *"If they (CHWs) were not there, it means you will be over worked, the clinic will always be full, to be honest. There is good communication between us." (Facility Manager, interview, Site 1)* However, even with a senior supervisor, if the team were located in a health post at a distance from the main clinic

(*Site 4*), the CHW struggled to engage with clinic staff, and to ensure patients accessed to care at the clinic. (For a full description of the supervisory practices, please refer to our companion paper.[16])

## Value for money

Across the district as a whole, the annual expenditure on the programme was R44 million (2016 Rands), at an average cost per CHW of R52 000 (2016 Rands). The cost per capita (for the district population without private health insurance) was R47. District expenditure per capita on primary healthcare (including community health centres and clinics, but not district hospitals) was R1200. We therefore estimated that WBOT expenditure in the district was only 3.9% of PHC expenditure.

The costs were higher in the models with: (1) start-up costs of the health post; (2) more supervisors and (3) a smaller number of CHW per supervisor (table 5). The EN-only model in the clinic (Sites 5 and 6) was the cheapest; although this model achieved higher coverage, the observation data suggests the care provided was poorer. The health post at some distance from the clinic (Site 4) was mid-range cost but as the nurses were trying to provide a basic clinic service, the CHW programme suffered. The health post close to the clinic was expensive because there were a small number of CHW, but the quality of care was good. The PN & EN model (Sites 1 and 2) in the clinic represented the best value for money as the team was well integrated into the health system, even if there was insufficient space in Site 2.

## DISCUSSION

In summary, household coverage was limited in all the sites (10%–20%). The low proportion of registration visits suggests that the CHWs were revisiting households they know (driven by the requests by the clinic to deliver medication or trace patients who had stopped attending the clinic) rather than taking on the more difficult task of approaching an unknown household, and so limiting the number of households they were responsible for. Insufficient training and resources demotivated the CHW and generated resentment. Our study suggests that the CHW could achieve a greater number of visits per day, and that more effort could be made towards targeting households in greater need (eg, those with young children). An increase to four visits per CHW pair per day would achieve coverage ranging from 30% to 90%, suggesting that some teams need more CHWs, as well as better supervision. Across the six sites, assuming that CHW work in pairs for security, they spend 4 days a week visiting households (other days will be used for campaigns, leave or compiling statistics), make four household visits a day (one household registration visit and three follow-up visits per day), a pair of CHW could care for approximately 220 households.

With the respect to quality of care, from the observations, we learnt CHWs did not ask all the registration questions; without knowing the full range of health and social needs of the households, they are unlikely to provide the necessary care. However, the CHW identified additional need(s) (beyond the purpose of the visit) in a third of households. From the household survey, we learnt household respondents remembered between a fifth to a quarter of the health messages, and half of patients took the referral action recommended by the CHW.

With respect to supervision and location, where there was better, more senior supervision, household members were more likely to recall the advice and messages given; where there is less supervision and less training, the CHWs are visiting more households, but providing poorer quality care. Teams based in a clinic, with a senior supervisor, are more likely to be better integrated into the health system, and so able to provide a higher quality service. Without senior supervision, the EN supervised teams, although located in a clinic, were not integrated into the system as they did not receive the necessary training or ongoing support, and they struggled to negotiate effective working relationships with other facility staff members. In the health post that served a community without a clinic, nurse supervisors had to consult directly with patients and had less time to supervise the CHWs, and engagement with the clinic was difficult due to the distance.

Expenditure on the CHW programme is only 3.9% of expenditure on non-hospital primary healthcare. Greater investment in more capable supervision, and where necessary, a greater number of CHW, would increase coverage, and improve the quality of the service. Box 1 sets out our recommendations from this phase of the study. Given the shortage of professional nurses, scaling up the PN & EN model is not feasible, so cost-effective ways need to be found to train and provide ongoing to support to the EN supervisors in order to improve the quality of their supervision. In the second phase of this study, we are evaluating

---

**Box 1 Recommendations for an effective CHW programme**

1. Dedicated, experienced supervision (both to build CHWs' and their supervisor skills, and to facilitate collaboration with the closest facility) is essential otherwise the investment in CHW programme is unlikely to achieve the household coverage or quality of care possible.
2. Where there are insufficient numbers of CHW, the programme needs to comprise a larger proportion of the PHC budget, in order to substantially increase the coverage of the programme.
3. A room inside the clinic to meet and compile reports is a necessary but not sufficient for the integration between clinic and the CHW team. Where physically not possible, an additional structure should be build next to facility.
4. Health posts in locations without a formal clinic to serve the community should be converted into formal clinics, or at least provided with sufficient resources and infrastructure so that they can operate as a full clinic.

CHW, community health workers.

---

at an intervention to provide mentorship to the EN supervisors in Sites 5 and 6.

## Limitations

Due to the very low number of households reporting a CHW visit in the past month, analyses of specific services were not possible. As a result, we have relied on the observation of household registration visits to assess the comprehensiveness of the service provided. The number of messages received during the CHW's visit in the last month, reported by the household member, depended on their recall. During observations, the work effort by the CHW increased due to the presence of fieldworker; we observed the same CHW over 4 days in order to reduce this effect. However, we were also able to estimate the likely number of visits made per day from the household survey results. We were not able to do a cost–benefit analysis and have instead compared the costs to overall primary healthcare expenditure to assess affordability and value for money.

The study's strength is the wide range of data collection methods that enabled triangulation across the different data sources, including the observation of CHW activity rather than self-reported data. The six sites were typical of the Sedibeng District, and while it is not possible to generalise with any certainty, the sites included both rural and periurban communities typical of much of the country.

Sufficient number of CHW to achieve adequate coverage is clearly important; however, the optimal number of households per CHW depends on her/his role, as well as geographic characteristics of the area.[7 17] Most evaluations of CHW programmes are focused either on a single condition, or a cluster of conditions (eg, maternal and child health); there is limited research assessing the performance of CHW in more comprehensive programmes. Where delivery of care is focused on maternal and child health, CHWs have been found to struggle with providing comprehensive care, with gaps in the information provided by the CHW, due to insufficient supervision and support from the associated health service[18 19] In Rwanda, the addition of family planning to CHW's wide range of promotive and preventive tasks didn't place an undue burden on the CHW, although it was noted there were insufficient CHW to meet the family planning needs of their catchment area.[20] In response to the HIV epidemic, studies examining the expansion of CHW programmes to include ARV treatment found sustainable and consistent funding to be the key barrier.[21]

Effective linkages with the formal health system are key, including being perceived by other health workers as an integral and essential member of the health team, and as the foundation of the health system.[7] Several recent studies have focused on the degree of integration of CHW with the health system, identifying barriers such as resistance from other health workers, discrimination against CHW based on social, gender and economic status, ineffective incentives, inadequate supplies and infrastructure.[22–24] An evidence review found that in large-scale programmes,

supervision is almost always weak, with those responsible for supervision frequently having other responsibilities (such as patient care at a peripheral facility), and they often have no specific training in supervision of CHW.[25] Weak management and organisational structures lead to poor quality work, low morale and absenteeism.[26] The transition to comprehensive care in the Western Cape Province in South Africa was hampered by weak support systems (supervision, monitoring, financing, training) as well as insufficient subdistrict capacity for planning and management.[10]

There is little research on both household coverage and the quality of care provided; this is important as there is a potential trade-off between the two. In addition, there is little on evidence the synergistic effect of senior supervision, which may be important in enabling integration with the health system, and improving both coverage and quality of care. Given the shortage of human resources of health in low-income and middle-income countries, innovative strategies to strengthen supervision strategies are required.[7 27]

## CONCLUSION

The CHWs provided their service to less than a fifth of households in the catchment communities. The service the CHWs provided to the households they visited was limited. Where skilled senior supervision was available, the CHWs were able to provide a service that covered a broader range of health needs. Support from the local facility for the CHW team is key to providing a quality service, and a senior supervisor can help to facilitate this collaboration. Greater investment in numbers of CHW, supervisors, training and equipment is required for the potential benefits of the programme to be delivered.

**Author affiliations**
[1]Centre for Health Policy, Faculty of Health Sciences, University of the Witwatersrand, Johannesburg, South Africa
[2]School of Public Health, Faculty of Health Sciences, University of the Witwatersrand, Johannesburg, South Africa
[3]Medical School, University of Warwick, Warwick, UK
[4]Health Systems, South African Medical Research Council, Tygerberg, South Africa

**Acknowledgements** We would like to thank all of the study participants, as well as district and provincial management for their support and engagement with the study.

**Contributors** JG, MT and FG conceptualised the study and raising the funding. JG led the drafting of the manuscript. JG, JDK, OB and HM were responsible for overseeing data collection. ED was responsible for costing both data collection and analysis. MM, JL, JDK and OB were responsible for statistical analysis. JG, FG, HM, YT, TR, NN and JAW were responsible for qualitative analysis. JG, FG, HM, YT, TR, NN, JAW, MT and OB contributed to the drafting the manuscript. All authors have read the final draft.

**Funding** The study was funded by Medical Research Council UK, DFID, ESRC and Wellcome Trust under the Joint Health Systems Research Initiative.

**Competing interests** None declared.

**Patient consent for publication** Not required.

**Ethics approval** The study was approved by the University of the Witwatersrand's Human Research Ethics Committee (Medical) (M160354), the Gauteng Provincial

Health Research Committee and the Biomedical and Scientific Research Ethics Committee (BSREC) at University of Warwick (REGO-2016-1825).

**Provenance and peer review** Not commissioned; externally peer reviewed.

**Data availability statement** Data are available on reasonable request. Quantitative data can be made available on request. Given the difficult of anonymising qualitative, we will work with any researchers wishing to further analyse the qualitative data.

**ORCID iDs**
Jane Goudge http://orcid.org/0000-0001-6555-7510
Frances Griffiths http://orcid.org/0000-0002-4173-1438

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
