## [Reviewer comments · BMJ Open]

ARTICLE DETAILS

TITLE (PROVISIONAL)	Household coverage, quality and costs of care provided by community health worker teams and the determining factors: Findings from a mixed methods study in South Africa
AUTHORS	Goudge, Jane; de Kadt, Julia; Babalola, Olukemi; Muteba, Michel; Tseng, Yu-hwei; Malatji, Hlologelo; Rwafa, Teurai; Nxumalo, Nonhlanhla; Levin, Jonathan; Thorogood, Margaret; Daviaud, Emmanuelle; Watkins, Jocelyn; Griffiths, Frances

VERSION 1 – REVIEW

REVIEWER	Jane Njeru Mayo Clinic, USA
REVIEW RETURNED	16-Dec-2019

GENERAL COMMENTS	General: This study seeks to address important questions, and uses mixed methodology appropriately. There are several areas that need to be improved, as detailed below. In general, this paper needs a thorough review to correct grammatical errors across most of the sections in the paper. Multiple abbreviations are used throughout this paper, and it was not always easy to locate the full words (e.g MUAC on page 6, line 1). The full meaning of any abbreviations should be noted immediately after the first time that abbreviation is used the document. Abbreviations should be fully spelt out in the footnotes of the tables as well. Abstract: The abstract is a nice summary of the work, but needs some edits: - The methods subsection needs to spell out in more detail each of the methods employed - The last sentence in the results section is an extrapolation of the results, not part of the results themselves, and is also not included in the rest of the text. The authors could consider placing this in the conclusions subsection. Introduction: The introduction provides a good summary of the effectiveness of community health worker programs. -The authors identify the challenge of finding appropriate supervision for community health workers, and how this influences the location of programs. Is this a problem unique to South Africa, or is it more generalized? Please provide citations to support these statements. -The 2nd paragraph of the introduction describes the CHW program in South Africa. However, the timeline of when this program was initiated is missing; this would be helpful in contextualizing this study and the findings. Since the proportion of households visited by a CHW is an important outcome for this type, if there are specific
---

	expectations, this should be noted as well. Methods: Household survey: 220 households were selected per site (please note site 3 had to 24). There are several statements that need to be clarified further: - How were potential participants invited to participate in the survey? - Was survey collection done in person or electronically? - Please expound on the last statement under the household survey subsection, with better description of the logistic regression methodology, and variables considered. Qualitative work: - It is difficult to follow the exact methodology of the qualitative work, and it would be helpful to address the handling of each (observations, focus group discussions and interviews), followed by how the team combined the data thus obtained to draw the conclusions noted. Results: The first subsection under the results is really a description of the community health worker characteristics (training [please expound on the term 'matric' page 5, line 53], length of employment) programs, clinics, facilities, equipment and supervisors. It is a fairly long subsection, and was difficult to read through. The authors could consider presenting this data in the form of a table with succinct descriptions of each site across the characteristics above. This way, it would be easier for the reader to note any significant differences between the sites. Is there support to use the dwelling type as a proxy for social economic status (page 6, line 44)? If so, please add appropriate citations. Quotations from focus group discussions are identified as such, and one assumes that the other quotations are from interviews. If this is the case, please identify them as such. 1225 households were surveyed (check totals on the tables). Use of the term " individuals surveyed" in table 3, is confusing, as it gives the impression that each of those individuals was surveyed (including several individuals age 0-4). However, from the rest of the paper, the impression is that households were surveyed, and it may be more proper to term this as "total number of individuals in surveyed households". Please clarify if this is the case. In tables 3 and 4 with total number of individuals or household per section would be helpful. Discussion: Nice discussion on the findings. As noted under introduction above, providing more detailed background information on this community health worker program would be helpful. For example, what is the expected household coverage per community health worker in a given period of time? The second paragraph of the introduction brings out some important points. The community health worker identified needs beyond the purpose of the visit in the 3rd of household. Did these self-identified
--	--

	needs correlate with what the household respondents were more likely to remember? The authors have described the cost of the programs. However, this would make more sense in the context of cost-benefit analysis. The benefit analysis in this study is at best observational, since actual outcomes of this program in relation to the described goals (provide health promotion, prevention, screening and referral for a wide range of health conditions, page 2, lines 33-34), are not given. This should be acknowledged as an important limitation, likely beyond the scope of this work.
--	---

REVIEWER	Louis Jenkins Stellenbosch University South Africa
REVIEW RETURNED	22-Dec-2019

GENERAL COMMENTS	The article is well written, clear, scientific, and the Methods are explained well. The research question fills a scientific and social knowledge gap.
--

REVIEWER	Meghan Kumar Liverpool School of Tropical Medicine UK
REVIEW RETURNED	04-Mar-2020

GENERAL COMMENTS	Dear authors - thanks for this contribution to the literature around CHW supervision and performance. I appreciate the mixed methods approach to the problem and the comparison of different approaches, as well as the recognition of the variation between individuals in the supervision system. I have also read the companion paper and appreciate learning from your work! I would suggest several revisions to improve the clarity and impact of this paper. I will first structure these in relation to the checklist and then add further comments on specific sections:  1. Point 1: clarifying the research objective with a clear question or aim in the abstract would be helpful. How do you 'assess' the various aspects? Or what is the benchmark? 2. Point 2: The methods section is insufficient in the abstract. Add at least one sentence to clarify the qualitative and quantitative approaches; this is already there in the body of the paper. Perhaps less background and more methods in abstract, recognizing length limitations? 3. Point 4: Suggest including discussion guide / costing tools as supplementary material for replication purposes. Really interesting approach to integrating the different findings; requires judgement and the expertise of this strong team of authors. This is not reflected in your results section and is an opportunity. 4. Point 6: In abstract, you state that you will assess: "coverage, quality of care, and costs of the service provided by CHW teams with differing configurations of supervisors". These 'tool-oriented' outcomes are very clear in the methods – what isn't clear is the outcomes related to value for money and how you came to those. 5. Point 8: I would expect to see at least 30+ references for a paper of this complexity and length – particularly in the discussion section as well as perhaps to justify your choice of methods? 6. Point 9: See below comments on results section – pull the VfM
--

	findings out of the table and discuss; integrate across tools if possible – the Table does a good job of it but the text not yet reflecting the same. 7. Point 12: Limitations are only provided in the summary at the outset – you could come back and address what you think the impact of those is on the findings you observed in the discussion. 8. Abstract: You don't do justice to the range of methods you have used and the results you have obtained. Less background/conclusions and more on methodology and results here (esp qualitative and costing?) would be helpful to give the reader a sense of the breadth of the work and findings. See also comments on 'integrated' findings under the results below. 9. Results: Unfortunately the strong description of integrating the findings in the methods section is not reflected in the results section (except to some extent in the table) – consider having each subheading could reflect a key integrated finding rather than a methodological approach, as the latter seems to have led you to 'silo' the findings by tool. This is where I would focus your efforts in the revision. 10. Discussion: a. Please ensure that the first sentence of each paragraph is a topic sentence guiding the reader through – strengthening the logical flow here will help you make a compelling argument supported by the efforts put in to get high quality mixed methods data. b. More references in the discussion would show better how this work links to existing findings, again to place it better in the vibrant discussions around community health, performance, quality, etc.
--	---

VERSION 1 – AUTHOR RESPONSE

Reviewer(s)' Comments to Author:

Reviewer: 1

Reviewer Name: Jane Njeru

Institution and Country: Mayo Clinic, USA

Please state any competing interests or state 'None declared': None declared

General:

This study seeks to address important questions, and uses mixed methodology appropriately. There are several areas that need to be improved, as detailed below. In general, this paper needs a thorough review to correct grammatical errors across most of the sections in the paper.

- We have read through the paper to identify and correct the grammatical errors and hope we have identified them all.

Multiple abbreviations are used throughout this paper, and it was not always easy to locate the full words (e.g MUAC on page 6, line 1). The full meaning of any abbreviations should be noted immediately after the first time that abbreviation is used the document. Abbreviations should be fully spelt out in the footnotes of the tables as well.

- We have carefully read through the paper to ensure that all acronyms and abbreviations are spelt out in full at first mention, including the tables; we realize there are several abbreviations, however, we hope now that the abbreviations no longer hamper the reader's understanding.

Abstract:

The abstract is a nice summary of the work, but needs some edits:

The methods subsection needs to spell out in more detail each of the methods employed

- We have listed the specific methods used in the methods section of the abstract. As these methods are quite standard, and the limited number of words in the abstract, we haven't described the details of each method. Please let us know if you disagree with this decision. It now reads:

"We used mixed methods (a random household survey, focus group discussions, interviews and observations of CHW at work) to examine the performance of six CHW teams in vulnerable communities in Sedibeng, South Africa"

The last sentence in the results section is an extrapolation of the results, not part of the results themselves, and is also not included in the rest of the text. The authors could consider placing this in the conclusion subsection.

- We changed the last sentence from: "if CHW carried out four visits a day.." to "We estimated that if CHW carried out four visits a day, coverage would increase to 30-90% of households..." We hope that this change conveys that these figures were actually calculated from our data. We have decided to leave it in the results section of the abstract. We are happy to move it to the conclusion if this is more appropriate for the editorial style of the journal.

Introduction:

The introduction provides a good summary of the effectiveness of community health worker programs.

The authors identify the challenge of finding appropriate supervision for community health workers, and how this influences the location of programs. Is this a problem unique to South Africa, or is it more generalized? Please provide citations to support these statements.

- This problem is not unique to South Africa. The shortage of health care professionals, particularly of professional nurses, is well established in LMICs. We have referenced the WHO's 2006 World Health Report that describes the extent of the shortage of human resources for health. We have also referenced the synthesis of evidence on community health care workers conducted by John Hopkins School of Public Health, that concludes that the lack of appropriate supervisors is common.

The 2nd paragraph of the introduction describes the CHW program in South Africa. However, the timeline of when this program was initiated is missing; this would be helpful in contextualizing this study and the findings. Since the proportion of households visited by a CHW is an important outcome for this type, if there are specific expectations, this should be noted as well.

- We have added these two pieces of information and this section now reads as follows:

"South Africa initiated a national CHW programme (called ward-based outreach teams) in 2011 to strengthen primary health care.[9] The intention is to provide health promotion, prevention, screening services and referral for a wide range of health conditions. [10] While policy documents suggested that CHW should care for approximately 150-250 households,[11] it is not clear how many households the CHW are able to serve, or whether they are able cope with the broader range of health needs they are meant to provide services for."

Methods:

Household survey: 220 households were selected per site (please note site 3 had to 24).

- Please note 224 households were interviewed in site 3. See table 3

There are several statements that need to be clarified further:

How were potential participants invited to participate in the survey?

- We have expanded on the section, and the text now reads as follows:

"Fieldworkers used a random walk and a specified skip pattern in a designated area to select households. The household was approached and the member who knew most about the health of

other members, was invited to participate in the survey. Their responses were recorded on an electronic device.”

Was survey collection done in person or electronically?

- The survey was done in person. This becomes clear with the revisions in the sentence above.

Please expound on the last statement under the household survey subsection, with better description of the logistic regression methodology, and variables considered.

- We have included a more detailed description of the regression methodology, including the following text:

“A logistic regression was conducted to better understand who was receiving CHW services. We included the following variables: model type, ratio of households per CHW pair, whether the household has a person over 60, a child under 5, number of health care needs, distance from the facility, and dwelling type (as a proxy for SES). We first conducted an unadjusted analysis to determine whether any of the variables were significantly associated ($p < 0.2$) with receiving a CHW visit. All significant explanatory predictors were included in the model, with those that were not significant ($p > 0.05$) being removed one by one, with the least significant being removed first. The analysis took into account the effect of clustering due to sampling and levels of strata by using a robust standard-error estimates for stratified sampling.”

Qualitative work:

It is difficult to follow the exact methodology of the qualitative work, and it would be helpful to address the handling of each (observations, focus group discussions and interviews), followed by how the team combined the data thus obtained to draw the conclusions noted.

- As suggested, we separated the section on qualitative data collection into three separate parts (each with their own heading) in order to explain our handling of each methodology. These sections read as follows. This is followed by the section explaining how we combined the data.

"Qualitative data collection

Trained fieldworkers were given an orientation to community-based health care by an experienced primary health care nurse. Interview guides and observation templates were revised after internal piloting and feedback from fieldworkers. Respondents, except for CHW, were chosen purposively.

Focus group discussions

All CHW at work on the day of the focus group discussion were included. A brief survey captured key demographic and career history of the CHW. The topics discussed included descriptions of the types of activities carried out by CHW and the challenges they faced.

Observation of CHW

The selection of CHW to be observed was done randomly (drawing names out of hat) on the first morning of a 4-day observation period. The fieldworkers observed the CHW with or without supervisor at work and took detailed notes in a template.

Interviews

Interviews were conducted with the facility manager, clinic staff, and CHW supervisors to discuss the typical activities of the CHW, how the programme ran, and its successes and challenges. If a client was given a referral by a CHW, the fieldworkers asked to conduct an interview with the client in a month's time. This subsequent interview provided information on follow up actions taken by the client.

"

Results:

The first subsection under the results is really a description of the community health worker characteristics (training [please expound on the term 'matric' page 5, line 53], length of employment) programs, clinics, facilities, equipment and supervisors. It is a fairly long subsection, and was difficult to read through. The authors could consider presenting this data in the form of a table with succinct descriptions of each site across the characteristics above. This way, it would be easier for the reader to note any significant differences between the sites.

- We have put the information into table 3 as suggested. We removed the word 'matric', and instead inserted the following text:

"the percentage who had completed their high school education ranged from 25% to 63%."

Is there support to use the dwelling type as a proxy for social economic status (page 6, line 44)? If so, please add appropriate citations.

- In South Africa, even with a democratic government, the geography of apartheid still exists. This means that socio-economic status is quite visible in dwelling types, with squatter camps, townships and more formal areas. As a result, housing is a relatively good proxy for socio-economic status. This is true for other low and middle income settings. We have added some references in the methods section to support the use of housing as a proxy of socioeconomic status.

Quotations from focus group discussions are identified as such, and one assumes that the other quotations are from interviews. If this is the case, please identify them as such.

- We have labelled relevant quotes as being from interviews.

1225 households were surveyed (check totals on the tables). Use of the term "individuals surveyed" in table 3, is confusing, as it gives the impression that each of those individuals was surveyed (including several individuals age 0-4). However, from the rest of the paper, the impression is that households were surveyed, and it may be more proper to term this as "total number of individuals in surveyed households". Please clarify if this is the case.

- This is the case (households were surveyed). We have changed the heading in the table to read INDIVIDUALS IN HOUSEHOLDS SURVEYED

In tables 3 and 4 with total number of individuals or household per section would be helpful.

- The total number of households and individuals is presented in Table 3 and 4. It is the top row of each section, in bold

Discussion:

Nice discussion on the findings.

As noted under introduction above, providing more detailed background information on this community health worker program would be helpful. For example, what is the expected household coverage per community health worker in a given period of time?

- We have included text in the introduction to explain that the policy documents suggest that a CHW should care for 150-250 households. (Please see response above). We have also added the following text:

Across the 6 sites, assuming that CHW work in pairs for security, spend 4 days a week visiting households, make 4 household visits a day (1 household registration visit and 3 follow up visits per day), a pair of CHW could care for approximately 220 households."

The second paragraph of the introduction brings out some important points. The community health worker identified needs beyond the purpose of the visit in a 3rd of households. Did these self-identified needs correlate with what the household respondents were more likely to remember?

- The data on whether the CHW asked all the registration questions, and whether the CHW identified any additional needs, came from the observed household visits. The data on the messages from the last months visit recalled by the respondents comes from the household survey. The observed

households were not necessarily included in the household survey, and so it is not possible to assess whether they are correlated. We have made it clearer that these two pieces of data are from different sources in the following paragraph:

“From the observations of CHW visits, we learnt CHW did not ask all the registration questions; without knowing the full range of health and social needs of the households, they are unlikely to provide the necessary care. However, the CHW identified additional need(s) (beyond the purpose of the visit) in a third of households. From the household survey, we learnt household respondents remembered between a fifth to a quarter of the health messages, and half of patients took the referral action recommended by the CHW.”

The authors have described the cost of the programs. However, this would make more sense in the context of cost-benefit analysis. The benefit analysis in this study is at best observational, since actual outcomes of this program in relation to the described goals (provide health promotion, prevention, screening and referral for a wide range of health conditions, page 2, lines 33-34), are not given. This should be acknowledged as an important limitation, likely beyond the scope of this work.

- We have acknowledged this as a limitation in the following sentence: “We weren’t able to do a cost-benefit analysis and have instead compared the costs to overall primary health care expenditure to assess affordability and value for money.”

Reviewer: 2

Reviewer Name: Louis Jenkins

Institution and Country:

Stellenbosch University

South Africa

Please state any competing interests or state ‘None declared’: None declared

Please leave your comments for the authors below

The article is well written, clear, scientific, and the Methods are explained well.

The research question fills a scientific and social knowledge gap.

Reviewer: 3

Reviewer Name: Meghan Kumar

Institution and Country:

Liverpool School of Tropical Medicine

UK

Dear authors - thanks for this contribution to the literature around CHW supervision and performance. I appreciate the mixed methods approach to the problem and the comparison of different approaches, as well as the recognition of the variation between individuals in the supervision system. I have also read the companion paper and appreciate learning from your work!

I would suggest several revisions to improve the clarity and impact of this paper. I will first structure these in relation to the checklist and then add further comments on specific sections:

1. Point 1: clarifying the research objective with a clear question or aim in the abstract would be helpful. How do you 'assess' the various aspects? Or what is the benchmark?

- We have re-written the research objective in the abstract in the following way, defining explicitly what we mean by coverage.

In this paper, we assess coverage (proportion of households visited by a CHW in the past year & month), quality of care, and costs of the service provided by CHW teams with differing configurations of supervisors, some based in formal clinics and some in community health posts.

We haven't defined quality of care here in the abstract as we use two rather complex indicators, and we are of the view that this level of detail would be too overwhelming in the abstract. (The two indicators are: number of messages recalled by household respondent from a CHW visit in the last month, and proportion of pre-specified questions asked at household registration). However, we have defined these indicators in the methods. If you rather we put this level of detail in the abstract, we can do so.

2. Point 2: The methods section is insufficient in the abstract. Add at least one sentence to clarify the qualitative and quantitative approaches; this is already there in the body of the paper. Perhaps less background and more methods in abstract, recognizing length limitations?

- We have expanded the methods section in the abstract which now reads as follows. As these methods are quite standard, and the limited number of words in the abstract, we haven't described the details of each method. Please see response above.

We used mixed methods (random household survey, focus group discussions, interviews and observations of CHW at work) to examine the performance of six CHW teams in vulnerable communities in Sedibeng, South Africa

3. Point 4: Suggest including discussion guide / costing tools as supplementary material for replication purposes. Really interesting approach to integrating the different findings; requires judgement and the expertise of this strong team of authors. This is not reflected in your results section and is an opportunity.

- The topics covered in the focus group discussion were quite standard, and are listed in table 2. Hence we not convinced that providing the actual tool will add much value, but can do so if required. Re integration of the results please see our response to the comment below

4. Point 6: In abstract, you state that you will assess: "coverage, quality of care, and costs of the service provided by CHW teams with differing configurations of supervisors". These 'tool-oriented' outcomes are very clear in the methods – what isn't clear is the outcomes related to value for money and how you came to those.

- In comment 6 below the reviewer makes the point that the table does a good job of explaining how we reached the value for money judgments, but this needs to be reflected in the text. We have done this, as suggested. Please see response to point 6 below.

5. Point 8: I would expect to see at least 30+ references for a paper of this complexity and length – particularly in the discussion section as well as perhaps to justify your choice of methods?

- We have increased the number of references, in both the introduction and the discussion section, which has been considerably strengthened and expanded, drawing on a wider range of literature.

6. Point 9: See below comments on results section – pull the VfM findings out of the table and discuss; integrate across tools if possible – the Table does a good job of it but the text not yet reflecting the same.

- We have rewritten the text on the costs so that it more closely matches that in the table:

"The costs were higher with: a) the start-up costs of the health post; b) more supervisors, and c) a smaller number of CHW per supervisor (Table 5). The EN-only model in the clinic (Sites 5 & 6) was the cheapest; although this model achieved higher coverage, the observation data suggests the care provided was poorer. The health post at some distance from the clinic (Site 4) was mid-range cost but as the nurses were trying to provide a basic clinic service the CHW suffered. The health post close to

the clinic was expensive because there were a small number of CHW, but the quality of care was good. The PN & EN model (Site 1&2) in the clinic represented the best value for money as the team was well integrated into the health system, even if there was insufficient space in Site 2.”

7. Point 12: Limitations are only provided in the summary at the outset – you could come back and address what you think the impact of those is on the findings you observed in the discussion.

- We have included a specific section on the limitations of the paper, and have discussed the impact of those limitations:

"Limitations

Due to the very low number of households reporting a CHW visit in the past month, analyses of specific services were not possible. As a result, we have relied on the observation of household registration visits to assess the comprehensiveness of the service provided. The number of messages received in the CHW in the last month, reported by the household member, depended on their recall. During observations, the work effort by the CHW increased due to the presence of fieldworker; we observed the same CHW over 4 days in order to reduce this effect. However, we were also able to estimate the likely number of visits made per day from the household survey results. We weren't able to do a cost-benefit analysis and have instead compared the costs to overall primary health care expenditure to assess affordability. "

8. Abstract: You don't do justice to the range of methods you have used and the results you have obtained. Less background/conclusions and more on methodology and results here (esp qualitative and costing?) would be helpful to give the reader a sense of the breadth of the work and findings. See also comments on 'integrated' findings under the results below.

-As mentioned above have added more information on the methods used in the abstract. We have also deleted a sentence from each of the background and the conclusion to release extra words. We have responded to the point about integration of findings below.

9. Results: Unfortunately the strong description of integrating the findings in the methods section is not reflected in the results section (except to some extent in the table) – consider having each subheading could reflect a key integrated finding rather than a methodological approach, as the latter seems to have led you to 'silo' the findings by tool. This is where I would focus your efforts in the revision.

- Below we have listed the headings from the results section, and the methods used to collect the data in that section, under that heading. This allowed us to reflect on rationale for different sub-sections and the extent to which the sub-sections used data from the different methods.

The clinics, health posts and the CHW teams

The data used here comes from all of the data sources ie. interviews with the facility manager, the clinic staff, focus group discussions with CHW, mini questionnaire with the CHW, observations of household visits and CHW team and their supervisor at work, except the household

Household characteristics and need for care

The data used here did just come from the household survey. However, we think it is appropriate to have a section that describes the needs of the households in the catchment area

Household coverage

Again this data did come from the household survey. As coverage is a key outcome, again we would argue this needs to have its own section

Type, and quality of care, delivered

This section draws on all the data sources ie from observations of household visits, interviews with facility managers, supervisors, CHW, and CHW clients, as well as the household survey

Supervision practices and support from the health system

This section draws on all the data sources ie the observations of CHW and supervisor activities, household CHW visits, focus group discussions with CHW, and interviews with the facility manager, the supervisor and the clinic staff, except the household survey

Value for money

This section draws on all the methodological approaches in the study. It draws findings on coverage, quality of care and costs, in order to make judgments about value for money.

We would like to suggest that these headings reflect key characteristics of the programme, and although 2 of the sub-sections only draw on one source (the household survey), the choice of headings was not driven the underlying data sources, rather the key elements of the programme that the study set out to examine. We have changed the heading 'costs' to 'value for money' as this better reflects the fact that this section draws on the data reported in the other sub-sections, and so on all the methodological approaches.

10. Discussion:

a. Please ensure that the first sentence of each paragraph is a topic sentence guiding the reader through – strengthening the logical flow here will help you make a compelling argument supported by the efforts put in to get high quality mixed methods data.

-We have edited the discussion to that each paragraph starts with a topic sentence indicating the subject of the paragraph. The section on limitations (and strengths) has now has its own heading

Para 1: In summary, household coverage was limited in all the sites (10-20%).

Para 2: With the respect to quality of care, from the observations, we learnt CHW did not ask all the registration questions...

Para 3: With respect to supervision and location, where there was better, more senior supervision, household members were more likely to recall the advice and messages given; where there is less supervision and less training, the CHW are visiting more households, but providing poorer quality care.

Para 4: Expenditure on the CHW programme is only 3.9% of expenditure on non-hospital primary health care.

b. More references in the discussion would show better how this work links to existing findings, again to place it better in the vibrant discussions around community health, performance, quality, etc.

-We have considerably expanded the discussion section, drawing on a much broader range of literature. There are now 27 references.